# Modeling the kinetics of the neutralizing antibody response against SARS-CoV-2 variants after several administrations of Bnt162b2

**Quentin Clairon**[1,2,3☯¤], **Mélanie Prague**[1,2,3☯¤], **Delphine Planas**[3,4], **Timothée Bruel**[3,4‡],
**Laurent Hocqueloux**[5‡], **Thierry Prazuck**[5‡], **Olivier Schwartz**[3,4‡],
**Rodolphe Thiébaut**[1,2,3‡]*, **Jérémie Guedj**[6]

**1** Université de Bordeaux, Inria Bordeaux Sud-Ouest, Bordeaux, France, **2** Inserm, Bordeaux Population Health Research Center, SISTM Team, UMR1219, Bordeaux, France, **3** Vaccine Research Institute, Créteil, France, **4** Virus and Immunity Unit, Institut Pasteur, Université de Paris Cité, CNRS UMR3569, Paris, France, **5** Service des Maladies Infectieuses et Tropicales, Centre Hospitalier Régional, Orléans, France, **6** Université Paris Cité, IAME, Inserm, Paris, France

☯ These authors contributed equally to this work.
¤ Current address: Bordeaux Population Health Research Center, Université de Bordeaux, 146 rue Léo Saignat,33076 Bordeaux cedex, France
‡ TB, LH, TP and OS also contributed equally to this work.
* rodolphe.thiebaut@u-bordeaux.fr

**Data Availability Statement:** All relevant data are within the manuscript and its Supporting information files.

## Abstract

Because SARS-CoV-2 constantly mutates to escape from the immune response, there is a reduction of neutralizing capacity of antibodies initially targeting the historical strain against emerging Variants of Concern (VoC)s. That is why the measure of the protection conferred by vaccination cannot solely rely on the antibody levels, but also requires to measure their neutralization capacity. Here we used a mathematical model to follow the humoral response in 26 individuals that received up to three vaccination doses of Bnt162b2 vaccine, and for whom both anti-S IgG and neutralization capacity was measured longitudinally against all main VoCs. Our model could identify two independent mechanisms that led to a marked increase in measured humoral response over the successive vaccination doses. In addition to the already known increase in IgG levels after each dose, we identified that the neutralization capacity was significantly increased after the third vaccine administration against all VoCs, despite large inter-individual variability. Consequently, the model projects that the mean duration of detectable neutralizing capacity against non-Omicron VoC is between 348 days (Beta variant, 95% Prediction Intervals PI [307; 389]) and 587 days (Alpha variant, 95% PI [537; 636]). Despite the low neutralization levels after three doses, the mean duration of detectable neutralizing capacity against Omicron variants varies between 173 days (BA.5 variant, 95% PI [142; 200]) and 256 days (BA.1 variant, 95% PI [227; 286]). Our model shows the benefit of incorporating the neutralization capacity in the follow-up of patients to better inform on their level of protection against the different SARS-CoV-2 variants.

**Funding:** Work in OS lab is funded by Institut Pasteur, Urgence COVID-19 Fundraising Campaign of Institut Pasteur, Fondation pour la Recherche Médicale (FRM), ANRS, the Vaccine Research Institute (ANR-10-LABX-77), Labex IBEID (ANR-10-LABX-62-IBEID), ANR / FRM Flash Covid PROTEO-SARS-CoV-2, ANR Coronamito, HERA european funding, Sanofi and IDISCOVR. This work has received funding from the French Agency for Research on AIDS and Emerging Infectious Diseases via the EMERGEN project (ANRS0151). This work was supported by INSERM and the Investissements d'Avenir program, Vaccine Research Institute (VRI), managed by the ANR under reference ANR-10-LABX-77-01. The funders had no role in study design, data collection and analysis, decision to publish, or preparation of the manuscript.

**Competing interests:** The authors declare no competing interests.

**Trial registration**: This clinical trial is registered with ClinicalTrials.gov, Trial IDs NCT04750720 and NCT05315583.

## Author summary

Developed vaccines against SARS-CoV-2 have been a turning point against the ongoing Covid-19 pandemic. When the Wuhan virus was dominant, they help to dramatically reduce the number of severe cases as well as infection and transmission rates. For mRNA vaccines, it was in great part explained by the high level of induced antibodies a few weeks/months after injection and linked to high neutralizing capacity, the ability to prevent viruses to enter and infect target cells. However, decreasing antibody concentration over time and apparition of variants escaping their neutralizing action dramatically reduced the initial vaccine efficacy. As a countermeasure, additional injections were used to re-establish significant antibody population and ensure a long-term neutralizing activity against emerging variants. To infer if this multi-dose strategy fulfills such task, we construct a model of the evolution of the induced antibodies and their neutralizing capacity against different variants. This model helps us to quantify the gain brought by each new injection on both antibody population and their neutralizing ability against all tested variants as well as the dramatic differences between them. We also predict the long-term evolution of neutralizing activity, years after last injection, and thus discuss the longevity of the induced protection by vaccine.

## Introduction

The discovery and the rapid availability of several vaccines against SARS-CoV-2 has been a turning point in the combat against Covid-19 [1]. Although their efficacy may vary to some extent, it is undisputable that large scale vaccination campaigns have dramatically reduced both the risk of severe diseases [2–4] and, to a lesser extent, the rates of transmission and disease acquisition [5–7], resulting in millions of saved lives [1, 8, 9].

However vaccine efficacy has been jeopardized by the apparition of various Variants of Concern (VoCs) that partially escape immune protection. A clear decrease in the neutralization capacity has been observed [10, 11] which has translated to a substantial reduction of efficacy against transmission and disease acquisition with Delta and Omicron variants, and, to a lesser extent, to a decrease of efficacy against severe Covid-19 disease [12, 13]. The concern caused by a potential loss of protection against VoCs has been further enhanced by the natural waning immunity and the progressive reduction in antibody levels over time [14–16]. This has supported boosting strategies with one or two additional vaccine doses to maintain a high level of protection. However the optimal time to administer boosters, and how these times may vary for different VoCs, remains unclear.

To characterize in detail the duration of protection against SARS-CoV-2, it is therefore essential to measure not only total anti-S IgG antibodies over time, as typically done in large observational studies, but also how this translates in terms of neutralization capacity. The latter requires intensive *in vitro* measurements, but it provides a much more accurate description of the level of protection present in the sera of Covid-19 vaccine recipients [17, 18]. Then, a detailed characterization of the immunological or virological factors modulating the

duration of protection can be obtained by using mathematical models of immune marker dynamics [19].

Here we propose to use for the first time a mathematical model to analyze the joint kinetics of anti-S IgG antibodies and neutralization capacity after repeated vaccine injections against the main VoCs. For that purpose we relied on data from a cohort of Bnt162b2 vaccine recipients, in which both antibody kinetics and neutralizing activity were measured longitudinally [11, 20, 21]. We built on previous models of antibody kinetics [22, 23] to develop a novel approach to quantify the kinetics of neutralizing activity, and we use this model to characterize the effects of repeated vaccine administrations on it. We finally use the model to discuss the duration of protection conferred by the measured humoral activity induced by Bnt162b2 against VoCs.

## Materials and methods

### Ethics statement

This study was approved by the Ethics Committee ILE DE FRANCE IV. The cohort was approved by the national external committee (CPP Ile-de-France- IV IRB No. 00003835). Study participants did not receive any compensation. At enrolment a written informed consent was collected for all participants.

### Data

**Population study.**   Data originate from a cohort of N = 29 subjects who received up to three injections of Bnt162b2 (ClinicalTrials.gov:NCT04750720 and ClinicalTrials.gov: NCT05315583). In brief SARS-CoV-2 naive patients were recruited in Orléans, France between August 27, 2020 and May 24, 2022. Individuals were followed for up to 483 days after their first vaccine injections (see more details on the data in [11, 20, 21]). Two patients without longitudinal follow-up and 1 immunocompromised individual were not included in our analysis. In total, N = 26 individuals were analyzed (see Table 1). Briefly, all subjects received at least 2 doses, administered on average 27 days after the first injection. N = 22 subjects received a third injection, administered on average 269 days after the first injection. During the follow-up N = 12 had a positive PCR, and only data prior to infection were analyzed, leaving an average follow-up of 11 visits and a median follow-up time of 362 days.

**Longitudinal markers of immune response.**   Two types of measurements were available at each visit: 1) anti-spike binding IgGs, measured in BAU/mL) neutralization titers of sera

**Table 1. Characteristics of the analyzed population.**

| Characteristics | Median | Median Time of vaccination | |
|---|---|---|---|
| | [Min; Max] or <u>n</u> (%) | [Min; Max] | |
| | | since first dose | since second dose |
| Men | 14 (54%) | | |
| Age | 59 [33; 95] | | |
| Follow-up duration after first-dose (days) | 368 [168; 483] | | |
| Number of follow-up visits | 14 [2; 18] | | |
| Number of vaccination doses | | | |
| 1st | 26 (100%) | - | - |
| 2nd | 26 (100%) | 22 [17; 60] | - |
| 3nd | 23 (88%) | 243 [175; 385] | 221 [154; 361] |

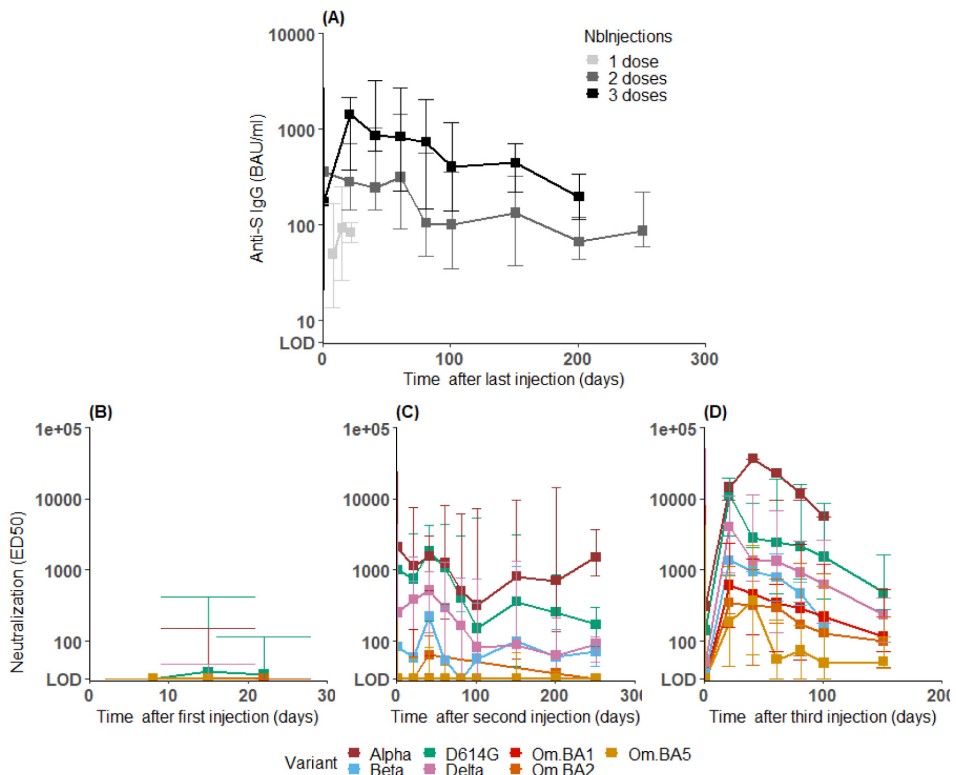

**Fig 1.** A: longitudinal evolution of the binding antibody concentration of anti-S IgG. B-C-D: longitudinal evolution of the neutralizing activity against VoCs after the first (B, see S1 Fig for a zoomed version), second (C) and third (D) vaccination dose. Squares represent median values, and plain horizontal lines represent the minimal and maximal encountered values among subjects. The lower limit of detection (LOD) is equal to 6 BAU/mL for IgG and 30 for $ED_{50}$. Given the limited number of samples available, data were grouped, using a one week sliding window after the first dose, 20 days in the first 100 days following the second or third infection, and 50 days for the other data points.

provided in $ED_{50}$, which is the effective dilution required to neutralize 50% of an arbitrary viral load of reference (eg, the higher the $ED_{50}$ the larger the protection level). Neutralization capacity was assessed against historical strain (D614G), Alpha, Beta, Delta, and Omicron variants (strains BA.1, BA.2, and BA.5).

In brief IgGs markedly increased after each dose, but rapidly declined over time, with a rate that did not substantially differ after the second or the third dose (Fig 1A). In contrast the kinetics of neutralizing activity was much more heterogeneous, and was characterized by large differences against the different VoCs. Further the neutralizing activity was markedly increased after the third dose against all Omicron variants, albeit remaining at much lower levels than against the other VoCs (Fig 1B, 1C and 1D).

## Model for neutralizing antibody response

**Mechanistic model for antibody kinetics.**   We rely on a simplified and rescaled version of a previously published model in the context of vaccine against Ebola infection [23]. In brief, after each dose, cells transfected with Bnt162b2 generate antigen, noted *V*, which triggers the constitution of a memory compartment, noted *M*, at a rate $\rho$. This memory compartment is a general one accounting for all cell populations able to differentiate into secreting cells upon antigen presence. These can be activated or memory B-cells either circulating or present in

germinal centers. So, $M$ can differentiate into secreting plasma cells, noted $\tilde{S}$, at a rate $\mu V$. These cells then produce antibodies, noted $Ab$, at a rate $\theta$. $V$, $S$, and $Ab$ are degraded at rates $\delta_V$, $\delta_S$, and $\delta_{Ab}$, respectively, leading to the following ODE system:

$$\dot{V} = -\delta_V V$$
$$\dot{M} = \rho V - \mu VM$$
$$\dot{\tilde{S}} = \mu VM - \delta_S \tilde{S}$$
$$\dot{Ab} = \theta \tilde{S} - \delta_{Ab} Ab.$$

(1)

Assuming that individuals are naive of infection, and noting $\mathbf{t_1}$ the time of first injection, the initial conditions are given by: $M(\mathbf{t_1}) = \tilde{S}(\mathbf{t_1}) = Ab(\mathbf{t_1}) = 0$.

To model the effect of repeated doses, we consider that $V$ is a function presenting discontinuity at time of first, second and third injection ($\mathbf{t_1}$, $\mathbf{t_2}$, $\mathbf{t_3}$). By denoting $k = 1, \ldots, 3$ the dose number, on each interval $[\mathbf{t_k}, \mathbf{t_{k+1}}]$, solving previous ODE for $V$ gives us $V_k(t) = V_0 e^{-\delta_V(t-\mathbf{t_k})}$ where $V_0$ is the initial antigen concentration, assumed equal from all doses.

Because this model is not identifiable when only $Ab$ are measured, we derived a structurally identifiable approximated model described in Eq 2; see Appendix A.1 in S1 Appendix for a description of this simplification. Briefly, it consists of rescaling the model for $S = (\mu V_0 \overline{M_1})^{-1} \tilde{S}$ and assuming that $M$ can be replaced by its steady-state value $\overline{M_k}$ if equilibrium is reached quickly after each injection:

$$\dot{S} = f_{\overline{M_k}} e^{-\delta_V(t-\mathbf{t_k})} - \delta_S S$$
$$\dot{Ab} = \vartheta S - \delta_{Ab} Ab$$

(2)

where $f_{\overline{M_k}} = \frac{\overline{M_k}}{\overline{M_1}}$ is the fold-change for steady-state memory compartment after $k^{th}$ injection compared to the first one (by definition $f_{\overline{M_1}} = 1$). Of note, we also tested the full model which does not assume a steady state value for M. This leads to identifiability issues mainly due to $\mu_S$ estimation for which only a lower bound ($\mu_S > 20$) can be found. For such values for $\mu_S$, the compartment $M$ nearly instantaneously reaches its steady-state. Accordingly, both full and simplified models provide virtually similar predictions for $Ab$ (see Appendix A.2 in S1 Appendix). Finally, we also tested a more complex model accounting for a delay between vaccine injection and antibody production (also in Appendix A.2 in S1 Appendix). However the model did not improve data description, which was probably due to the limited amount of information available on antibody kinetics in the couple of days following vaccine injection. Moreover, the model proposed by Balelli et al. [23] initially contains two populations of secreting cells $S$ and $L$, differing by their life expectancy. In our case, preliminary statistical analysis conclude that there was no statistical differences between model adjustments when accounting for $S$ and $L$ or $S$ only (results not shown). This allows us to reduce the number of unknown parameters. This is crucial for parameters related to cell kinetics known to be very different for newly developed mRNA vaccines comparing to viral vector ones and for which no values have been previously inferred. Thus, the retained model (2) is complex enough to account for the effect of multiple injections on antibody concentration evolution while avoiding identifiability issues. We define $\eta^{ODE} = (f_{\overline{M_2}}, f_{\overline{M_3}}, \delta_V, \delta_S, \vartheta, \delta_{Ab})$ the vector of model parameters defining the dynamics of the system.

**Functional model for neutralizing activity.** After modeling antibody concentration evolution in the previous section, we aim to model their neutralizing activity. This means in our

case proposing a model describing the evolution of $ED_{50}^v$ with respect to $Ab$. We consider the following linear model:

$$ED_{50}^v(t) = F(v, t)Ab(t).$$

The function $F(v, t)$ represents the relationship between the concentration of binding antibodies in BAU/mL and its neutralization capacity against the VoC $v$. It is variant-specific and time-varying, let us first derive its expression for the strain D614G before moving to any arbitrary VoCs. After $\mathbf{t_1}$, we assume a proportional relationship between $Ab$ and neutralizing activity against D614G i.e $F(D614G, t) = \gamma$ (equivalently $ED_{50}^{D614G}(t) = \gamma Ab(t)$). After additional injections, we assume there is a neutralization gain quantified by the fold-change $f_2$ after $\mathbf{t_2}$ and $f_3$ after $\mathbf{t_3}$ i.e. $F(D614G, t) = \gamma f_2$ when $t \in [\mathbf{t_2};\mathbf{t_3}]$ and $F(D614G, t) = \gamma f_3$ for $t \geq \mathbf{t_3}$. Now, we account for VoCs specific neutralizing activity by modifying baseline value $\gamma$ by the fold-changes $f_v$ such that $F(v, t) = F(D614G, t)f_v = \gamma f_v$ when $t \in [\mathbf{t_1};\mathbf{t_2}]$ and $F(v, t) = F(D614G, t)f_v = \gamma f_v f_2$ for $t \in [\mathbf{t_2};\mathbf{t_3}]$. We assume that the relative gain brought by third injection can be also VoC-specific. That is why we introduce the fold-changes $g_v$ to quantify this gain i.e. $F(v, t) = F(D614G, t)f_v g_v = \gamma f_v f_3 g_v$ for $t \geq \mathbf{t_3}$. This piece-wise constant function can be then expressed in a general form:

$$F(v, t) = \gamma f_v (\mathbb{1}_{t < \mathbf{t_2}} + f_2 \mathbb{1}_{t \in [\mathbf{t_2};\mathbf{t_3}]} + f_3 g_v \mathbb{1}_{t \geq \mathbf{t_3}}).$$

The choice of this model is the result of exploration based on the minimization of an adjustment criteria. In particular, the current model only quantifies the effect of the repetition of injections on affinity enhancement. Other factors can play a role as the elapsed time since antigen presentation, for example to account for the progressive Memory B-cells repertoire expansion [24, 25]. An alternative neutralization model only considering the time factor has been developed. This supplementary analysis is described in Appendix B in S1 Appendix but lead to a less accurate model (in terms of AIC). A general model accounting for both factors, the number of injections and the elapsed time, has been also tested leading to non-significant improvements over the retained model and at the expense of identifiability problems (results not shown). We also investigate the possibility of a variant-specific fold-change after second injection. This was discarded due to practical identifiability issues. More generally, our model assumes a simple linear relationship between antibody concentration and neutralization, with no saturation effect. The fact that a more physiological model assuming a nonlinear relationship did not improve data description (see Appendix B in S1 Appendix) suggests that the level of Antibody observed in this study remains within the linear range of neutralization. Finally, we acknowledge the existence of other ways than our descriptive approach to link $Ab$ and $ED_{50}^v$, as in Padmanabhan et al. [26] in which a mechanistic relationship between these quantities is constructed. Still, their model definition involves measurements, such as infection events or neutralizing antibodies, which are not at our disposal, especially for emerging VoCs, making their model intractable for our prediction purpose.

We define $\eta = (\eta^{ODE}, \gamma, f_v, f_2, f_3, g_v)$ the vector of model parameters that have to be estimated from the observed data. Description of the model parameters can be found in Table 2.

**Observation model.**   The structural model used to describe the log-transformed concentration of binding antibodies in BAU/mL for the $i^{th}$ individual ($i = 1, \ldots, N$) at the $j^{th}$ time point ($j = 1, \ldots, n_i$) is:

$$Y_{ij}^{BAU} = \log_{10}(Ab(\eta_i, t_{ij})) + e_{ij}^{BAU},$$

where $e_{ij}^{BAU}$ is the residual additive error which follows a normal distribution of mean zero and constant standard deviation $\sigma_{BAU}$. The vector $\eta_i$ is the specific value for individual $i$ of vector $\eta$.

**Table 2. Model parameters and estimation.** Fcn:fold-change in neutralization.

| Parameter | Description | Unit | Fixed Effect [IC95%] | Random Effect [IC95%] |
|---|---|---|---|---|
| $f_{\overline{M_2}}$ | Fold change for $M$ equilibrium after second injection | dimensionless | 7.1 [4.2; 12.0] | 0.9 [0.8; 1.0] |
| $f_{\overline{M_3}}$ | Fold change for $M$ equilibrium after third injection | dimensionless | 18.5 [15.0; 26.0] | |
| $\vartheta$ | Initial acceleration for $Ab$ production | $[A].days^{-2}$ | 24.5 [15.8; 38.0] | 0.5 [0.3; 0.7] |
| $\delta_{Ab}$ | Antibody degradation rate | $days^{-1}$ | 0.08 [0.07; 0.09] | |
| $\gamma$ | Proportion of neutralization provided by first vaccination | $[V].[A]^{-1}$ | 0.3 [0.2; 0.5] | 0.7 [0.5; 0.9] |
| $f_{Alpha}$ | Fcn for variant Alpha | unitless | 1.3 [1.0; 1.8] | |
| $f_{Bseta}$ | Fcn for variant Beta | unitless | 0.2 [0.1; 0.3] | |
| $f_{Delta}$ | Fcn for variant Delta | unitless | 0.3 [0.2; 0.4] | |
| $f_{BA.1}$ | Fcn for variant BA.1 | unitless | 0.005 [0.003; 0.009] | |
| $f_{BA.2}$ | Fcn for variant BA.2 | unitless | 0.013 [0.005; 0.029] | |
| $f_{BA.5}$ | Fcn for variant BA.5 | unitless | 0.016 [0.011; 0.022] | |
| $f_2$ | Fcn for second vs. first injection | unitless | 8.2 [4.0; 16.9] | |
| $f_3$ | Fcn for third vs. first injection in original strains D614G | unitless | 18.8 [10.0; 42.9] | |
| $g_{Alpha}$ | Fcn for third vs. first injection in variant Alpha | unitless | 5.8 [3.0; 11.8] | |
| $g_{Beta}$ | Fcn for third vs. first injection in variant Beta | unitless | 2.3 [1.3; 4.2] | |
| $g_{Delta}$ | Fcn for third vs. first injection in variant Delta | unitless | 1.1 [0.6; 1.5] | |
| $g_{BA.1}$ | Fcn for third vs. first injection in variant BA.1 | unitless | 13.5 [7.5; 24.3] | |
| $g_{BA.2}$ | Fcn for third vs. first injection in variant BA.2 | unitless | 5.4 [2.5; 11.9] | |
| $g_{BA.5}$ | Fcn for third vs. first injection in variant BA.5 | unitless | 1.7 [1.2; 2.5] | |
| $\sigma_{BAU}$ | Measurement error for $Y^{BAU} = \log_{10}(Ab) + e^{BAU}$ | | 0.24 [0.23; 0.25] | |
| $\sigma_{D614G}$ | Measurement error for $Y^{D614G} = \log_{10}(ED_{50}^{D614G}) + e^{D614G}$ | | 0.47 [0.44; 0.50] | |
| $\sigma_{Alpha}$ | Measurement error for $Y^{Alpha} = \log_{10}(ED_{50}^{Alpha}) + e^{Alpha}$ | | 0.59 [0.53; 0.64] | |
| $\sigma_{Beta}$ | Measurement error for $Y^{Beta} = \log_{10}(ED_{50}^{Beta}) + e^{Beta}$ | | 0.47 [0.41; 0.53] | |
| $\sigma_{Delta}$ | Measurement error for $Y^{Delta} = \log_{10}(ED_{50}^{Delta}) + e^{Delta}$ | | 0.42 [0.40; 0.44] | |
| $\sigma_{BA.1}$ | Measurement error for $Y^{BA.1} = \log_{10}(ED_{50}^{BA.1}) + e^{BA.1}$ | | 0.44 [0.36; 0.52] | |
| $\sigma_{BA.2}$ | Measurement error for $Y^{BA.2} = \log_{10}(ED_{50}^{BA.2}) + e^{BA.2}$ | | 0.48 [0.40; 0.56] | |
| $\sigma_{BA.5}$ | Measurement error for $Y^{BA.5} = \log_{10}(ED_{50}^{BA.5}) + e^{BA.5}$ | | 0.34 [0.30; 0.40] | |
| $\delta_V$ | Induced vaccine antigen declining rate | $days^{-1}$ | 2.7 (fixed) | |
| $\delta_S$ | Death rate of S cells | $days^{-1}$ | 0.01 (fixed) | |

We also consider a log-transformation of $ED_{50}^v$ raw measurements for the variant in the list {*D614G*, *Alpha*, *Beta*, *Delta*, *BA*.1, *BA*.2, *BA*.5}. For the $i^{th}$ individual at the $j^{th}$ time point, we have:

$$Y_{ij}^v = \log_{10}(ED_{50}^v(\eta_i, t_{ij})) + e_{ij}^v,$$

where $e_{ij}^v$ is the residual additive error for variant $v$ which follows a normal distribution of mean zero and constant standard deviation $\sigma_v$.

**Statistical model for parameters over time and injections.   Fixed parameters.** Here, not all parameters can be jointly estimated via likelihood when only concentration of binding antibodies and antibody neutralizing activity are measured. Further, the model predictions were found largely insensitive to the choice of the degradation rate of $V$ and $S$. Using a profiled likelihood approach [27], we fixed their half-life to 0.25 and 51 days, respectively.

**Inter-individual variability.** In the vector $\eta$, some parameters have to be individual-specific to account for inter-individual variability. It is the case for $\psi_i = (\vartheta, f_{\overline{M_2}}, \gamma)$. We suppose it

follows a log-normal distribution such that:

$$\psi_i = \psi_0 exp(u_i),$$

where $\psi_0$ is the fixed effect and average mean value in the population. The vector $u_i$ is individual random effects, which follow a normal distribution of mean zero and standard deviation $\Omega$, and account for heterogeneity across individual. We assume that other parameters in vector $\eta$ except error measurements are also estimated in log-transformation and are common to all individuals in the population. Altogether, the vector of parameters to estimate is given by $\theta = (\eta, \Omega, \sigma_{BAU}, \sigma_v)$.

**Estimation procedure**. Parameters were estimated (and named $\hat{\theta}$ in the following) with the SAEM algorithm implemented in MONOLIX software version 2022R1 [28] allowing to handle left censored data [29]. Likelihood was estimated using the importance sampling method and standard error were obtained by asymptotic approximation and inversion of the Fisher Information Matrix. Graphical and statistical analyses were performed using R version 3.4.3.

**Simulation of long-term humoral response.** Next, we used the model to predict the long-term evolution of $Ab$ and $ED_{50}^v$ over time. To account for uncertainty in our predictions, we used a Monte-Carlo sampling method, where $K = 1000$ replicates of parameters values $\theta^{(k)}$ were sampled in the posterior distribution of the parameter estimates to derive 95% prediction intervals (PI) of the predicted trajectories.

Finally we used these predictions to calculate the time to reach a given threshold value. To take into account between-subjects variability, we added a second layer to our Monte-Carlo sampling method and we sampled $N = 100$ replicates in the population parameter distribution. We used these predictions to derive the probability of having a concentration of binding antibodies greater than given thresholds, in particular higher than 264 BAU/mL, which corresponds to the standard threshold of protection defined by Feng et al. [30] and adopted by WHO. The level of neutralizing activity has been identified as a correlate of protection for vaccine efficacy against the historical strain [31, 32]. However, to date, no threshold for $ED_{50}^v$ value ensuring protection has been isolated for D614G, let alone for the new VoCs. So, for a range of threshold values, we calculated the probability that the neutralizing activity against each VoC would be higher than these values over time, especially if this activity was still detectable at a given time. In this way, we can compare the longevity of neutralizing activity between VoCs even in the absence of a clear threshold of protection for each of them.

## Results

### Mechanistic model for humoral response

We first aimed to investigate whether there is a proportional relationship between the evolution of concentration of binding antibodies and its neutralization capacity. Fig 2 displays the observed relationship from data between antibody concentration and $ED_{50}^v$ for each VoC after each injection. First, we notice that these ratios are different for the variants. Then, we compared the evolution of these ratios with respect to the previous vaccination. In most cases, the ratios improved significantly, indicating an intrinsic gain in neutralization that cannot be explained by the variation in antibody concentration alone, justifying the need to quantify this phenomenon precisely. This is supported by the linear regressions of $ED_{50}^v$ on $Ab$ after each injection presented in Table 3 (made with the R package Censreg [33] to account for censored data). These regressions indicate an increased correlation between $ED_{50}^v$ and $Ab$ with respect to the injection numbers for most of VoCs (already pointed out by Goel et al. [34] for D614G and Beta).

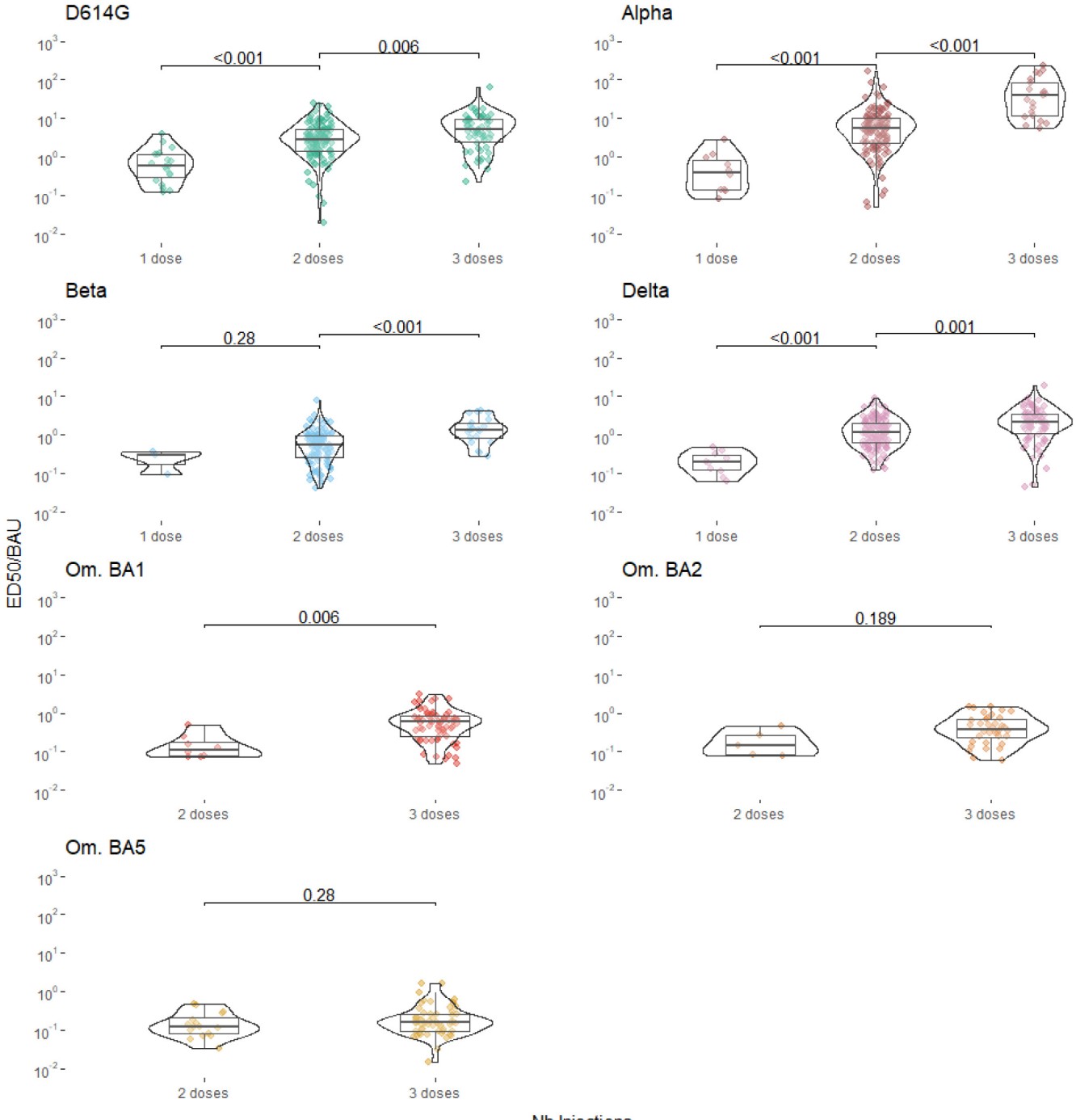

**Fig 2. Evolution of the predicted ratio $ED_{50}^v/BAU$ for each VoC after successive vaccine doses.** Each circle represents a ratio $ED_{50}^v/BAU$ computed when both measurements for $ED_{50}^v$ and $BAU$ where available for a given patient at a given observation time. Most of patients contribute several times due to the repeated measurements made over time after each dose. All predictions below the limit of detection for $ED_{50}^v$ were removed to avoid overoptimistic $ED_{50}^v/BAU$ ratio when replacing $ED_{50}^v$ values by detection threshold. This explains why very few values are available for Beta and Delta and none for Omicron strains for one dose case. Comparison between vaccine dose was done using Wilcoxon test with Holm correction, p-values are given above the brackets.

**Table 3. Linear regression results with censored data.**

| | $\beta$ estimation in model $ED_{50}^v = \alpha + \beta Ab$ on interval: | | |
|---|---|---|---|
| | **[t₁;t₃]** | **[t₂;t₃]** | **[t₃;+∞]** |
| D614G | 0.90 [0.01; 1.73] | 2.42 [1.48; 3.47] | 15.31 [8.93; 21.68] |
| Alpha | 0.50 [-0.26; 1.24] | 4.39 [2.02; 6.76] | 38.65 [3.24; 74.06] |
| Beta | 0.10 [-0.15; 0.30] | 0.30 [0.15; 0.45] | 1.83 [0.93; 2.70] |
| Delta | 2.05 [-10.5; 12.30] | 1.58 [1.11; 2.02] | 5.86 [4.16; 7.55] |
| BA.1 | - | 0.11 [0.01; 0.22] | 1.08 [0.74; 1.41] |
| BA.2 | - | 0.16 [0.02; 0.30] | 0.21 [0.08; 0.36] |
| BA.5 | - | 0.15 [0.07; 0.23] | 0.51 [0.31; 0.71] |

Estimation of model parameters can be found in Table 2. This estimation indicates that multiple injections both increase antibody concentration and intrinsic affinity per constant antibody unit. Regarding antibody concentration, estimation of mechanistic parameters indicates a significant increase in the size of the memory compartment. It increased by $f_{\overline{M_2}} = 7.1$ (95% Confidence interval CI [4.2; 12.0]) after the second injection and by $f_{\overline{M_3}} = 18.5$ (95% CI [15.0; 26.0]) after the third injection compared to the first one. Of note, the estimated value for $\delta_{Ab}$ approximately corresponds to an half-life of 9 days, which is close to the typical range of 10 and 21 days [35].

Regarding neutralization per constant antibody concentration unit, we found that there are two main influencing factors: the repetition of the injections and the VoC. Regarding repeated injections effect for the original strain D614G, the second dose increases neutralization by a factor $f_2 = 8.2$ (95% CI [4.0; 16.9]) and the third one by $f_3 = 18.8$ (95% CI [10.0; 42.9]) compared to the first injection. Now regarding the neutralization capacities for emerging VoCs, they are significantly decreased compared to the original strain, with the exception of Alpha, where there is no significant change in neutralization compared to D614G. It ranges from a reduction of 70% (95% CI [60%; 80%]) for Delta to a dramatic reduction of 99.5% (95% CI [99.1%; 99.7%]) for BA.1. Still, we find that the sequential injection strategy confers a gain in long-term neutralizing capacities for all VoCs. The second injection increases neutralization against all VoCs by the same factor $f_2 = 8.2$ (same as D614G). The third injection increases neutralization in a VoC-specific manner, given by $f_3 g_v$. It ranges from an increase in fold change of 21 (95% CI [6.0; 64.4]) for Delta to 254 (95% CI [75; 1042]) for BA.1 times higher for the third injection than for the first injection. For comparison with D614G, the neutralization is $\frac{f_3}{f_2} = 2.3$ (95% CI [1.6; 3.2]) times higher for the third injection compared to second injection. Transitively, the fold change is $\frac{f_3}{f_2} g_{Delta} = 2.5$ (95% CI [0.8; 4.8]) for Delta to $\frac{f_3}{f_2} g_{BA.1} = 31.1$ (95% CI [12.0; 77.8]) for BA.1 times higher for the third injection than for the second injection.

Examples of fitted trajectories are given for four randomly selected patients in Fig 3. We observe a very good adequation with most of the observations lying in the 95% prediction intervals. To assess the capability of the model to fit our data, we also examined the visual predictive check (see Appendix C in S1 Appendix), which showed that the model well captures the kinetics observed and its variability across individuals.

This is exemplified in Fig 4, that shows the mean markers trajectories for an average individual (i.e random effects $u_i$ set to 0). As expected, the level of the response is higher after a repeated number of injections for both binding antibody concentration and neutralization for all variants. Interestingly, the neutralization curves for BA.1, BA.2, and BA.5 are significantly

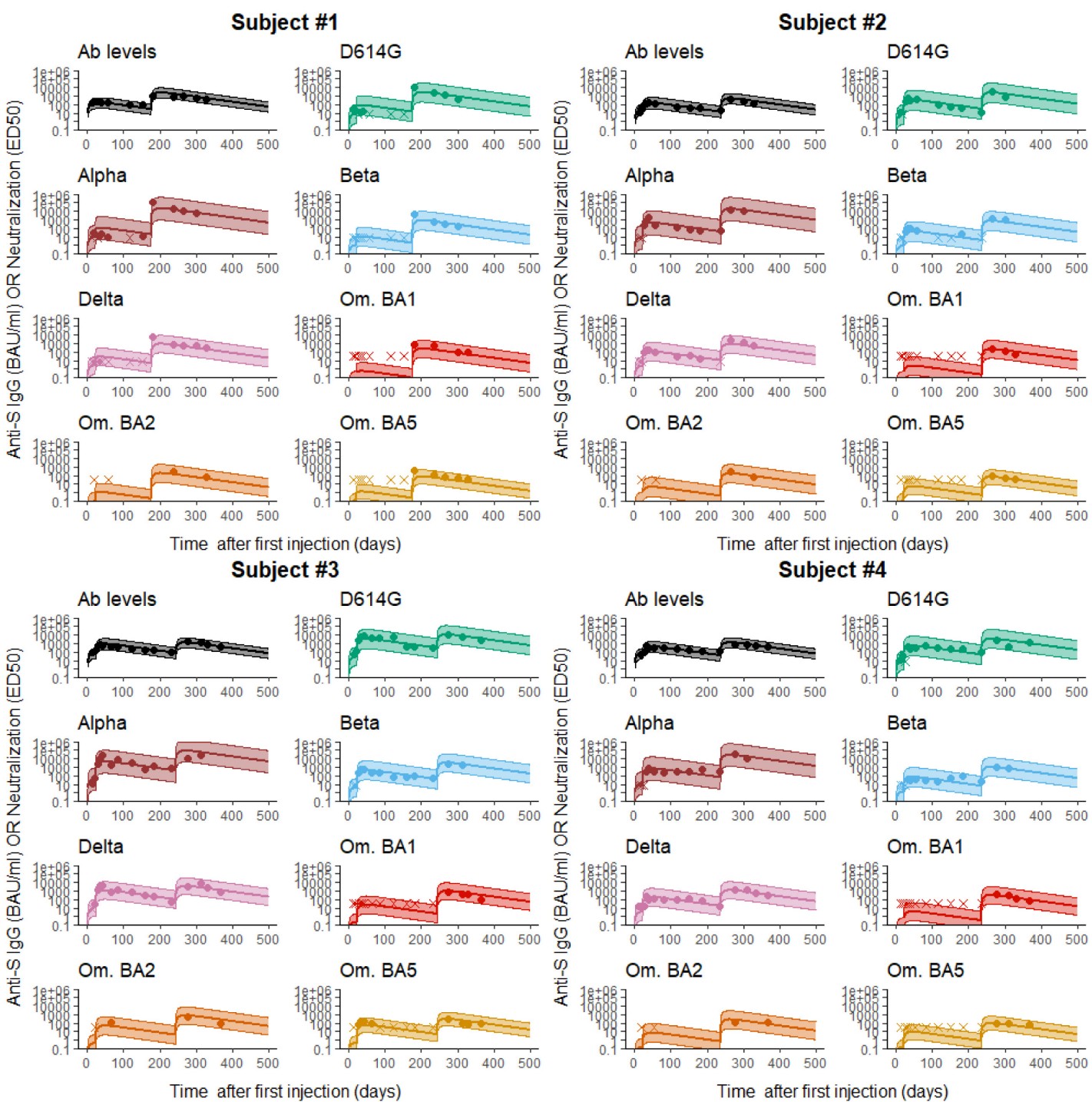

**Fig 3. Individual fits for four representative individuals.** The solid line is the subject-specific prediction and the shaded area is the 95% prediction interval. The plain dots and crosses represent the observed and censored data, respectively.

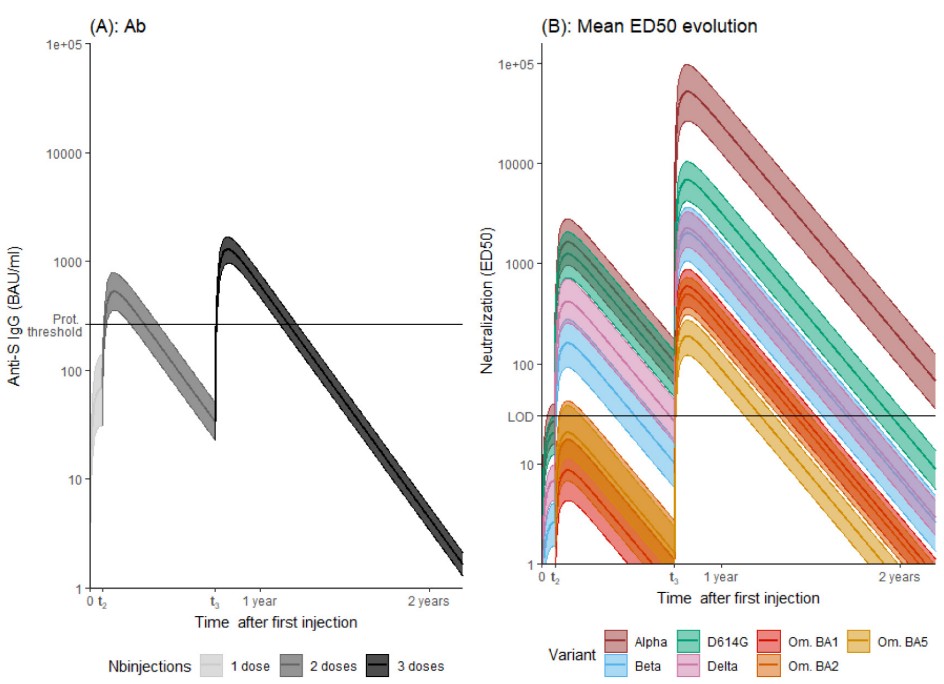

**Fig 4.** A: Predicted evolution of binding antibody concentration. The horizontal line corresponds to the value of 264 BAU/ml considered as a threshold against symptomatic infection. B: Predicted kinetics of $ED_{50}^v$. The horizontal line corresponds to the LOD. In all panels, the shaded area is the 95% prediction interval.

lower than for the other variants, with no overlap in prediction intervals. The first and second doses elicit a neutralization response for Omicron (BA.1, BA.2, and BA.5) that remains below the detection limit in most individuals (which is consistent with the observed data) but is dramatically enhanced by the third injection. Regarding the $ED_{50}^v/BAU$ ratio (See S2 Fig), we find that for the same concentration of binding antibodies, neutralization is significantly increased after each new injections for all variants and is significantly different for Alpha, D614G, {Delta, Beta} and {BA.1, BA.2, BA.5} variants.

## Long-term predictions

As already shown in Fig 4, we can use the estimated models to predict the long-term trajectories of markers corresponding to the mean parameter values as well as 95% prediction intervals. It allows to derive an estimation of the time needed to reach a certain threshold after a three injections vaccination scheme with first vaccination at time $t_1 = 0$, second injection at time $t_2 = 27$ days and third injection at time $t_3 = 269$ days, corresponding to the mean observed time of injection in our cohort. Binding antibodies concentration is below 264 BAU/mL 154 (95/% PI [137; 173]) days after third vaccination. Neutralization reaches undetectable levels between 173 days (95/% PI [142; 200]) for BA.5 to 587 (95/% PI [537; 636]) for Alpha after the third dose.

Fig 5A displays the probability of having antibody concentration higher than the protection threshold established by Feng et al. [30] of 264 BAU/mL each days after the last injection in the counterfactual scenario where subjects only received one, two or three doses. The same is done for neutralizing activity again the VoCs (Fig 5B: one, Fig 5C: two or Fig 5D: three). It is possible to see the drastic effect of repeated injections on the levels reached by both binding

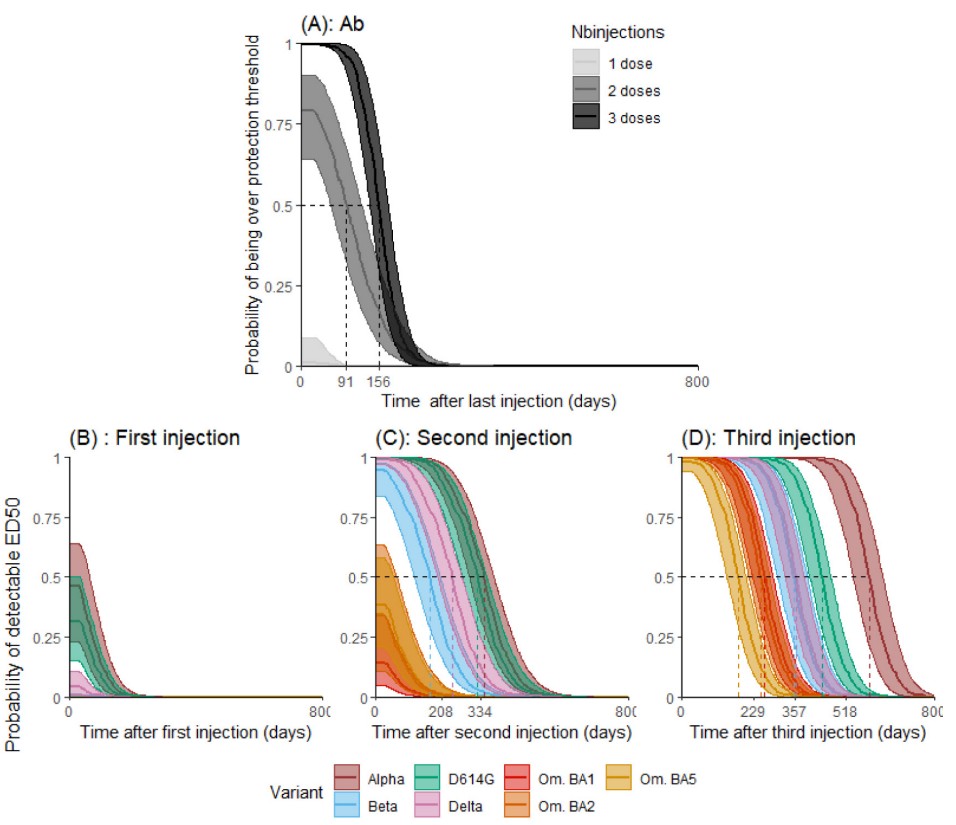

**Fig 5. A:** Predicted probability of having predicted antibody concentration (anti-S IgG) greater than 264 BAU/mL. B-C-D: Probability of having detectable neutralizing activity against VoCs after after the first (B), second (C) or third (D) vaccination dose. Simulations were performed assuming that the second and third vaccination doses occurred at day 27 and 269, respectively.

antibodies concentration and neutralization for all variants. Strikingly, the full response duration is similar in length for the binding antibodies concentration after two or three injections. However, whereas response higher than 264 BAU/mL is reached in 100% (95% PI [99%; 100%]) of the population after three injections, it is only reached in 82% (95% PI [65%; 90%]) of the population after two doses, and never reached in the whole population after the first injection (value: 0% (95% PI [0%; 3%])). Table 4 provides the time needed for a proportion of a vaccinated population to return under a certain threshold. It explores multiple thresholds (100 BAU/mL, 264 BAU/mL, and 1000 BAU/mL for antibodies concentrations; and undetectability, 100 and 1000 for neutralization) that could be investigated when and if a clear level of correlate of protection is found. For all markers, there is a systematically and significantly higher duration of humoral activity after three compared to two injections. After three injections, duration of neutralization against Omicron variants (BA.1,BA.2 and BA.5) is significantly lower than for other variants for all thresholds.

## Discussion

We proposed here a modeling framework to characterize the kinetics of antibodies to successive doses of Bnt162b2 vaccine. The originality of our approach is that we relied on both the kinetics of anti-S IgG binding antibodies and their neutralization against the major VoCs that

**Table 4. Predicted distribution for the duration of anti-S IgG and neutralization activity above different threshold levels [95% prediction interval].**

| | | Time to anti-S IgG | | |
|---|---|---|---|---|
| | Population quantiles | <100 BAU/mL | <264 BAU/mL | <1000 BAU/mL |
| IgG | 95% | 163 [146; 185] | 94 [74; 114] | 0 [0; 0] |
| | 50% | 223 [207; 242] | 152 [135; 173] | 62 [32; 76] |
| | 5% | 289 [265; 305] | 214 [193; 235] | 119 [97; 138] |
| | | Time to ED50 | | |
| Variant | Population quantiles | Undetectable | <100 | <1000 |
| D614G | 95% | 327 [294; 370] | 252 [207; 282] | 81 [39; 115] |
| | 50% | 433 [404; 471] | 350 [316; 382] | 186 [148; 217] |
| | 5% | 557 [506; 579] | 460 [420; 491] | 278 [252; 328] |
| Alpha | 95% | 476 [431; 528] | 401 [341; 447] | 230 [172; 283] |
| | 50% | 583 [539; 634] | 499 [445; 551] | 335 [280; 383] |
| | 5% | 706 [641; 745] | 609 [550; 657] | 427 [382; 491] |
| Beta | 95% | 237 [194; 290] | 161 [108; 199] | 0 [0; 0] |
| | 50% | 343 [301; 388] | 259 [217; 299] | 96 [44; 140] |
| | 5% | 467 [407; 499] | 369 [323; 405] | 188 [151; 247] |
| Delta | 95% | 245 [219; 289] | 170 [132; 195] | 0 [0; 0] |
| | 50% | 351 [323; 387] | 267 [238; 297] | 104 [67; 128] |
| | 5% | 475 [427; 497] | 377 [342; 407] | 196 [169; 241] |
| BA.1 | 95% | 146 [117; 183] | 70 [0; 97] | 0 [0; 0] |
| | 50% | 252 [222; 287] | 168 [139; 199] | 0 [0; 0] |
| | 5% | 376 [324; 398] | 278 [242; 309] | 97 [69; 141] |
| BA.2 | 95% | 132 [98; 181] | 56 [0; 89] | 0 [0; 0] |
| | 50% | 238 [203; 285] | 155 [119; 191] | 0 [0; 0] |
| | 5% | 362 [307; 392] | 265 [223; 301] | 83 [46; 131] |
| BA.5 | 95% | 62 [0; 99] | 0 [0; 0] | 0 [0; 0] |
| | 50% | 168 [143; 203] | 84 [54; 114] | 0 [0; 0] |
| | 5% | 292 [246; 310] | 194 [156; 225] | 0 [0; 52] |

have emerged since 2021. Our model quantifies the benefit of successive injections and can be used to predict the duration of detectable neutralizing activity against each VoC. After the first dose, the model shows the significant action of each additional injection, especially of the third one, to increase the intrinsic antibody neutralizing quality against all VoCs [34]. However, both the maximum level achieved and the rate of decline could vary greatly between VoCs. Accordingly, the mean duration of detectable neutralizing activity after the third dose of vaccine was 20, 12, 8.5, 8 and 6 months for Alpha, Delta, and Omicron BA.1, BA.2 and BA.5 respectively. Our results also highlight the wide variability in patient response, with at least 5% of patients with undetectable neutralizing activity against Omicron BA.5 only 2 months after the third injection.

These results were obtained based on a number of hypotheses, which we summarize below. First, the model of antibody concentration dynamics remains simplified, with the memory compartment simply represented by a piecewise constant function over successive doses. In addition, the model assumes only one type of secreting cell population and thus overlooks the complexity of the B-cell response mechanism. Our model does not integrate the possible mechanisms causing the gain in neutralization observed over dose injections, and how this may modulated by the time between injections. For instance, it has been suggested that longer delay between injections could increase the Memory B-cells selection stringency in germinal

centers [36]. This could in turn lead to strategies to maximize antibody concentration, as suggested by theoretical models [37]. This is in line with measurements made on another cohorts, for example the one described in [38, 39] with a significantly longer delay between injections comparing to ours. In this case, measured antibody concentration after third injection was significantly higher than our prediction. Still, due to our data limitation (few subjects with similar vaccination schedule), we cannot isolate and thus estimate the effect of injection schedule on neutralizing activity. The choice of linear relationship between neutralizing activity and binding antibodies obviously omits features acting on neutralization which could have been incorporated in a more complete model, closer to biological mechanisms. Still, S3 Fig shows that this choice is in adequation with general trend in our data and testing more complex relationships, such as a sigmoid model, did not lead to statistical improvement (see Appendix B in S1 Appendix). Of note, this linear relationship also provided a good fit to the data on external cohorts [40–42]. Finally, the model assumed that the second dose would result in a similar change in protection for all variants. In the future, application of such approaches to larger populations of individuals, with a wider range of tested vaccination schedules, may allow some of these hypotheses to be relaxed and injection timing to be integrated into the model specification without compromising the identifiability of the parameters.

Due to the available data and constructed model, we restrict our analysis to the humoral part of the immune response triggered by vaccination only. Still, vaccination also induces a cellular immune response which may contribute to the clinical protection especially against the VoCs [43] (for a mechanistic model accounting for T-cell response, see Korosec et al. [19]). Regarding immune response induced by infection, to this date, 11 out of 26 followed subjects were infected with Omicron. This proportion is likely to increase as it is the case in the global vaccinated population. Thus, it is of great interest to model the hybrid protection induced by vaccination followed by natural infection. Still, due to model limitation, we discarded patient data after breakthrough infection. It would requires to deeply modify our model to integrate two different antibody populations, one coming from vaccination and targeting the historical strain and the other one targeting the Omicron variant. That is why this analysis is left to future works.

One of the main advantages of the model is its flexibility to easily incorporate information on new VoCs and to use the strength of information obtained on other viral variants to update the model as data become available. In fact, we continuously updated the model to include successive Omicron variants. Interestingly, despite the small number of samples available, a high degree of precision was achieved for all variants. For example, although patients had on average only two data points with detectable neutralization against the BA.2 variant, this was sufficient to achieve a good precision for the estimation of the model parameters (Table 2). Also, despite its simplicity, the estimated mechanistic model for antibody kinetics produces consistent predictions for patients with a different vaccination schedule than the one considered here (see Appendix D in S1 Appendix). Additionally, by using all available data (eg, by analyzing anti-S IgG and neutralizing activity of all patients simultaneously), the model reveals some signals of kinetics that were not visible when analyzing the individual markers separately. For example, we identified different slopes of antibody decline that directly affect the prediction of protection duration. Using the same data set and a simpler single-slope model, the time to undetectable neutralizing levels after the third dose of vaccine for D416G was estimated to be 11.5 months [21], which is shorter than our estimate of 13.5 months (derived from Fig 4). In their approach, Planas et al. [21] chose to adjust the anti-S IgG and $ED_{50}^v$ decline separately for the different VoCs without considering causal relationships between them. On the contrary, our model assumes an influence of antibody concentration on the development of $ED_{50}^v$. Similar results are shown for the Delta variant (11.5 vs. 10.5 months, respectively), demonstrating

the importance of a model-based approach to predict neutralizing activity in the long term. Predictions for the Omicron strains were similar for BA.1 and BA.2, respectively 8.5 and 8 months, and show a reduction for BA.5 strains with 6 months. Interestingly, our results also suggest a longer duration of detectable neutralizing activity than what has been directly extrapolated from other observational studies [38, 44, 45], although this difference may also be due in part to the different experimental procedure used to measure neutralization.

The large and VoC-dependent variability in neutralization values argues for the use of individualized approaches to identify patients most at risk. Although it should be acknowledged that such an approach is hampered by the lack of an established neutralizing activity threshold as correlates of protection, its level was found to be associated with the risk of breakthrough infection. In a cohort of elderly nursing home residents, none of those the individuals with ED50 above 2136 had Omicron BA.1 breakthrough infection [46]. A model-based study found that a threshold of 1000 dramatically reduced peak viral load, suggesting that such a threshold may be a good indicator of protection against infection [40]. Interestingly, our results show that neutralizing levels for all Omicron variants remain largely below this value (Fig 5), consistent with the current understanding that BNT162b2 is poorly effective against disease acquisition in the Omicron era [39, 46]. Fortunately, the vaccine has shown high efficacy against severe disease to date [47, 48].

To date, the use of a fourth dose of vaccine to increase efficacy in France has been limited to high-risk patients who were not represented in this cohort. Nevertheless, we used the model to predict the neutralization levels that could be achieved after a fourth vaccine dose. Under the conservative assumption, yet consistent with available observational study [49], that this injection does not increase affinity or maturation parameters, our model predicts a similar duration of detectable neutralization as after the third dose, ranging from 172 to 256 days for the Omicron variants. Assuming that the fourth dose allows a similar increase in maturation and affinity as after the third dose, the model predicts that the duration of detectable neutralization could be much longer, ranging from 610 to 694 days for Omicron variants (see this supplementary analysis in Appendix E in S1 Appendix).

## Supporting information

**S1 Fig. $ED_{50}^{v}$ raw data after one injection.** Zoomed version of $ED_{50}^{v}$ raw data presentation after one injection.
(EPS)

**S2 Fig. Estimated mean evolution of $t \mapsto \frac{ED_{50}^{v}(t)}{Ab(t)}$.** Evolution of $ED_{50}^{v}$:BAU ratio.
(EPS)

**S3 Fig. Linear regressions $ED_{50}^{v} = \beta^{v} Ab$.** Linear regressions $ED_{50}^{v} = \beta^{v} Ab$ for each VoC from simultaneously measured neutralizing activity and binding antibody concentration. The censored data have been removed.
(EPS)

**S1 Appendix. Appendixes for "Modeling the evolution of the neutralizing antibody response against SARS-CoV-2 variants after several administrations of Bnt162b2".**
(PDF)

**S1 File. Available data.** The dataset used for this analysis is available in the zip file Neutralization_Data_and_code.
(ZIP)

## Acknowledgments

We thank Isabelle Staropoli, Florence Guivel-Benhassine, Françoise Porrot and all the members of the Virus and Immunity Unit for discussion and help, as well as Fabienne Peira, Vanessa Legros, Barbara De Dieuleveult, Aurelie Theillay, Sandra Pallay and Daniela Pires Roteia (CHR Orléans) for their help with the cohorts. Part of the experiments presented in this paper were carried out using the PlaFRIM experimental testbed, supported by Inria, CNRS (LABRI and IMB), Université de Bordeaux, Bordeaux INP and Conseil Régional d'Aquitaine (see https://www.plafrim.fr). We thank Simulations Plus, Lixoft division for the free academic use of the MonolixSuite.

## Author Contributions

**Conceptualization:** Quentin Clairon, Mélanie Prague, Rodolphe Thiébaut, Jérémie Guedj.

**Data curation:** Delphine Planas, Timothée Bruel, Laurent Hocqueloux, Thierry Prazuck, Olivier Schwartz.

**Formal analysis:** Quentin Clairon.

**Investigation:** Quentin Clairon, Mélanie Prague, Delphine Planas, Timothée Bruel, Laurent Hocqueloux, Thierry Prazuck, Olivier Schwartz, Rodolphe Thiébaut, Jérémie Guedj.

**Methodology:** Quentin Clairon, Mélanie Prague, Rodolphe Thiébaut, Jérémie Guedj.

**Resources:** Delphine Planas, Timothée Bruel, Laurent Hocqueloux, Thierry Prazuck, Olivier Schwartz.

**Software:** Quentin Clairon.

**Validation:** Quentin Clairon, Mélanie Prague, Rodolphe Thiébaut, Jérémie Guedj.

**Visualization:** Quentin Clairon.

**Writing – original draft:** Quentin Clairon, Mélanie Prague, Olivier Schwartz, Jérémie Guedj.

**Writing – review & editing:** Quentin Clairon, Mélanie Prague, Delphine Planas, Timothée Bruel, Laurent Hocqueloux, Thierry Prazuck, Olivier Schwartz, Rodolphe Thiébaut, Jérémie Guedj.

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
