## [Decision Letter · Decision Letter 0]

1 Feb 2023

Dear Dr Clairon,

Thank you very much for submitting your manuscript "Modeling the evolution of the neutralizing antibody response against SARS-CoV-2 variants after several administrations of Bnt162b2" for consideration at PLOS Computational Biology. As with all papers reviewed by the journal, your manuscript was reviewed by members of the editorial board and by several independent reviewers. The reviewers appreciated the attention to an important topic. Based on the reviews, we are likely to accept this manuscript for publication, providing that you modify the manuscript according to the review recommendations.

Sincerely,

James R. Faeder

Academic Editor

PLOS Computational Biology

Kiran Patil

Section Editor

PLOS Computational Biology

Reviewer's Responses to Questions

**Comments to the Authors:**

Reviewer #1: In this paper, Clairon et al. describe the coupling of a simple, mechanistic model of antibody production from B cells following mRNA vaccination against COVID-19 with a model of neutralization to understand the efficacy of mRNA COVID-19 vaccines against infection with respect to variants of concern. The authors use a simplified expression of their model to derive a piecewise description following multiple doses (primarily 3 doses of the orginal Wuhan strain vaccine, with some discussion of a 4th dose) of vaccine to understand the durability of neutralization over time. To parameterize their model, they use longitudinal data from a vaccine clinical trial in France and a non-linear mixed effects model (fit in Monolix). Their results indicate that third doses provide the highest neutralization and that Omicron is much less neutralized than other VOCs.

Major comments:

1) Understandably, the authors make use of the data that are available to them. In this case, this means a clinical trial where second doses were administered about 30 days after the first. However, certain countries (notably the UK, Canada, etc.) did not administer second doses according to this schedule, thereby affecting antibody titers in their populations over time. It would be interesting to use data from these countries as a validation/discovery substudy. What model parameters are altered in this case? Further, have the authors considered any validation beyond model fitting? This would help to convince on the predictability of the model beyond the individual time courses to which it is fit.

2) I have concerns about the model simplification used to address the model identifiability (and importantly, I believe to facilitate parameter estimation). As pointed out in the SI, the value \\bar{M_k} is the achieved equilibrium value. However, particularly for a low number of doses (i.e., 1-3), the behaviour of the model is still transient and thus how relevant or valid is it to use a steady-state-state assumption? Monolix allows for multiple doses so I'm not sure why this approach was used vs multiple doses in the dataset (one can see that the piecewise function was used for dosing in the attached code).

3) The mechanistic model of antibody production is quite simplified and relies on ODEs. However, it is known that there are delays in the processes of both generating antibody-producing B cells (here collapsed into one population M) and antibody from these cells. Since this model does not seem to take this into account, what are the implications of this modelling choice? It seems intuitively that they could be significant. This is somewhat discussed on page 5 line 95 but predictions are not shown and the discussion there does not relate specifically to this issue.

4) Further, the neutralization function used is linear instead of sigmoidal. In the SI, the authors discuss the AICs of each of their neutralization function hypotheses. Perhaps I've misunderstood their analysis, but why is the lowest AIC (H2) not chosen? Nonetheless, experimental data shows that patient-derived antibodies and monoclonal antibodies neutralize SARS-CoV-2 with a sigmoidal relationship and therefore the choice of neutralization function should be dictated by them and not a statistical argument, in my view.

5) The clearance rate of antibodies was fixed--couldn't this have a significant impact on the antibody kinetics being predicted?

Minor comments:

1) As the authors note in the discussion, effective neutralization and detectable neutralization (i.e., the difference between assay-based measurements of neutralization and real-world protection from infection) are different. Perhaps some care should be taken in the abstract with respect to the description of the findings.

2) There are a few typos throughout. Notably COVID-19 (all uppercase) and Covid-19 are both used and SARS-CoV-2 virus does not need the "virus" after the acronym. Adjustements should be Adjustments, for example.

Reviewer #2: This is a very nice paper on modeling the evolution of neutralizing antibody responses to SARS-CoV-2 variants following the administration of multiple doses of the Bnt162b2 vaccine. The paper simultaneously considers data of antibody concentrations (IgG levels) and their neutralizing ability (ED50) against all the major variants of concern following the administration of a single, two, and three doses of the vaccine. This is thus a rich and timely dataset, given the continued emergence of new variants. The authors develop a modeling framework that considers the dynamics of memory B cell formation and antibody production following vaccination. The data suggest that the IgG levels and ED50 levels are linked in non-trivial ways with the relationship between the two being different after each dose and with each variant. To recapitulate this observation, the authors use phenomenological constructs that link the IgG and ED50 levels after each dose and with each variants. They estimate these linking parameters using a mixed-effects modeling approach. The model seems to capture the antibody dynamics in individuals well. They then apply the model to predict the durability of the protective antibody response following one, two and three doses, for each of the variants. The predictions reinforce the prevalent notion that with some of the omicron variants the protection is not only limited but is also likely to wane soon after vaccination. The framework developed is useful and is likely to help analyze new data that may arise following additional vaccine doses or the emergence of new variants. The analysis is rigorous and the paper is well written. I have a few minor comments for the authors to consider.

Comments:

1. An important result from the study is that following the second or third dose, the neutralization capacity of antibodies ‘per capita’ increases compared to the previous dose (first or second) in most cases. This gives hope that with each subsequent dose, the protection will not only improve in quantity but also in quality, and thus become increasingly durable. While the model does well to recapitulate this observation using phenomenological constructs, the authors do not discuss possible mechanistic origins of this observation. A plausible explanation lies in the ongoing affinity maturation in germinal centres or the seeding of germinal centres with memory cells from previous doses. A simulation study explored these possibilities to explain the improvement in protection following delayed boosting (https://www.frontiersin.org/articles/10.3389/fimmu.2021.776933/full). The authors may wish to discuss this and any other hypotheses based on their data that they feel may offer some insights into the observation. Would, for instance, comparing the improvement in per capita neutralization ability following the third versus second dose be consistent with greater affinity maturation given the longer interval between the second and third doses compared to the first and second? This may also help understand why the function F linking ED50 and IgG levels must depend explicitly on time.

2. The authors cite the paper by Korosec et al (Ref. 19), but do not discuss the similarities or differences between their findings and those of the present study. I feel that this would be a useful discussion for the readers.

3. The authors focus on antibody responses and their neutralization capacity but may wish to comment on other arms of the immune response, particularly CD8 T cells, which may be triggered in response to vaccination. Although these may not have been significant against the ancestral strain, as the antibody efficacies drop against the variants, it could be that the relative roles of the cellular arms become significant. The authors may wish to comment on this possibility, especially since cellular responses as well as innate immune responses have been seen to Bnt162b2 vaccination (https://www.nature.com/articles/s41590-022-01163-9).

4. The authors construct a phenomenological relationship between antibody levels and ED50. In an interesting modeling study, a mechanistic description of this relationship for the ancestral strain has been developed (https://www.nature.com/articles/s43588-022-00198-0). The latter study also offers a potential interpretation of inter-patient variability. The authors may wish to mention this study, although its relevance to variants remains to be tested.

5. Minor comments:

a. The title indicates modeling the ‘evolution’ of antibody responses. The study, however, does not really model this evolution explicitly and so appears somewhat misleading.

b. The abstract mentions ‘optimal timing’ of vaccination. Given the phenomenological model for linking IgG titres with ED50, I am not certain how optimal timings would be arrived at. If this could be done, it would be worth including in the results.

c. The results of the fourth dose are in the supplementary and are discussed only at the very end of the discussion section. I wonder if these should be mentioned in the results section, especially since the fourth dose may be more widely administered in some countries like Israel than France.

Reviewer #3: In this manuscript, Clairon et al. use a mathematical model to understand the temporal dynamics of antibody titres (Ab) and neutralisation capacity against various SARS-CoV-2 variants of concerns (VOCs) following multiple doses of the BNT162b2 vaccine. They employ a model that was previously developed to examine vaccine-induced humoral responses against Ebola infection (PMID: 32205143). The central claim of this study is that the neutralisation capacity for a given antibody concentration increases after successive doses of vaccination. The authors then make predictions of long-term vaccine-induced antibody responses against SARS-CoV-2 VOCs. This study is interesting, timely, and well-conducted. However, there are some minor issues (see below) that would need to be addressed before acceptance.

(1) It would be helpful if the authors could provide more information on the statistical analysis in Figure 2:

1a. It needs to be clarified whether they have performed multiple comparison corrections (e.g., Bonferroni correction) and whether they report absolute or corrected p-values.

1b. It would help if the authors could mention the p-values rather than using asterisks.

1c. Also, the variance seems to be different across groups, and it is unclear if the data in some groups are normally distributed. It is therefore unclear whether the Student’s t-test is appropriate.

1d. I wonder if the authors could do a linear regression using the expression ED50 = gamma*Ab, estimate gamma at different doses, and show that gamma changes with doses. This might complement the parameter estimation using mixed-effect modelling performed in Figure 3.

(2) The authors could consider discussing their work in the context of existing COVID-19 vaccine modelling and experimental studies:

2a. A recent modelling study examined a plausible mechanistic link between antibody levels and ED50 for the original strain, and variations in ED50 amongst vaccinated individuals (DOI: 10.1038/s43588-022-00198-0).

2b. Simulation studies have explored how different vaccination protocols affect the quality and quantity of antibody responses following multiple vaccination doses (PMIDs: 34917089, 36353634).

2c. Goel et al investigated the relationship between antibody titre and FRNT50 (ED50 equivalent) for D614G and B.1.351 before and after booster. They found that the correlation between antibody titre and FRNT50 became stronger after booster (see Fig 1E in PMID: 33858945), suggesting an improvement in the quality of antibody response following booster dose.

(3) Minor issues:

3a. Figure 1 (bottom, left panel): It’s a bit hard to read the values. I wonder if providing a zoomed-in version as an inset would be helpful.

3b. Figure 1 (bottom, middle panel): The kinetics of ED50 does seem heterogeneous with time, as mentioned by the authors. Could this effect be simply due to data grouping?

3c. Ethics statement is a bit confusing because my understanding is that no new data is generated in this study, and all the human subject data is already published elsewhere (Refs. 20 and 21). Is this correct?

3d. Typo on line 96; ‘no’ instead of ‘not’?

**Have the authors made all data and (if applicable) computational code underlying the findings in their manuscript fully available?**

Reviewer #1: Yes

Reviewer #2: None

Reviewer #3: Yes

PLOS authors have the option to publish the peer review history of their article (what does this mean?). If published, this will include your full peer review and any attached files.

Reviewer #1: No

Reviewer #2: No

Reviewer #3: No

Figure Files:

Data Requirements:

Reproducibility:

References:

---

## [Decision Letter · Decision Letter 1]

15 May 2023

Dear Dr Clairon,

Thank you very much for submitting your manuscript "Modeling the kinetic of the neutralizing antibody response against SARS-CoV-2 variants after several administrations of Bnt162b2" for consideration at PLOS Computational Biology. As with all papers reviewed by the journal, your manuscript was reviewed by members of the editorial board and by several independent reviewers. The reviewers appreciated the attention to an important topic and two of the reviewers were satisfied that you addressed their concerns. We would ask you to kindly address the remaining concern of Reviewer 1. Based on the reviews, we are likely to accept this manuscript for publication, providing that you modify the manuscript according to the review recommendations.

Sincerely,

James R. Faeder

Academic Editor

PLOS Computational Biology

Kiran Patil

Section Editor

PLOS Computational Biology

Reviewer's Responses to Questions

**Comments to the Authors:**

Reviewer #1: I thank the authors for their revisions. Many of my previous questions were addressed but I still have an issue with the antibody neutralization function. The added statement that "Antibody observed in this study remains within the linear range of neutralization" can be tested through comparison with available antibody neutralization data. Why not test this explicitly through validation to external data?

I also found a few minor grammatical issues the authors may wish to correct:

-Revised title: kinetic should be kinetics

-p. 11: "Comparison between vaccine doses were" should be "Comparison between vaccine doses was"

Reviewer #2: The authors have addressed all my concerns. I have no further comments.

Reviewer #3: I thank the authors for addressing all my comments. I have no further suggestions/feedback.

**Have the authors made all data and (if applicable) computational code underlying the findings in their manuscript fully available?**

Reviewer #1: Yes

Reviewer #2: None

Reviewer #3: Yes

PLOS authors have the option to publish the peer review history of their article (what does this mean?). If published, this will include your full peer review and any attached files.

Reviewer #1: No

Reviewer #2: No

Reviewer #3: No

Figure Files:

Data Requirements:

Reproducibility:

References:

---

## [Editor Report · Decision Letter 2]

20 Jun 2023

Dear Dr Clairon,

We are pleased to inform you that your manuscript 'Modeling the kinetics of the neutralizing antibody response against SARS-CoV-2 variants after several administrations of Bnt162b2' has been provisionally accepted for publication in PLOS Computational Biology.

Best regards,

James R. Faeder

Academic Editor

PLOS Computational Biology

Kiran Patil

Section Editor

PLOS Computational Biology

---

## [Editor Report · Acceptance letter]

31 Jul 2023

PCOMPBIOL-D-22-01849R2 

Modeling the kinetics of the neutralizing antibody response against SARS-CoV-2 variants after several administrations of Bnt162b2

Dear Dr Clairon,

I am pleased to inform you that your manuscript has been formally accepted for publication in PLOS Computational Biology. Your manuscript is now with our production department and you will be notified of the publication date in due course.

With kind regards,

Zsofi Zombor
